# Human Rad51 Protein Requires Higher Concentrations of Calcium Ions for D-Loop Formation than for Oligonucleotide Strand Exchange

**DOI:** 10.3390/ijms25073633

**Published:** 2024-03-24

**Authors:** Axelle Renodon-Corniere, Tsutomu Mikawa, Naoyuki Kuwabara, Kentaro Ito, Dmitri Levitsky, Hiroshi Iwasaki, Masayuki Takahashi

**Affiliations:** 1Nantes Université, CNRS, US2B, UMR 6286, F-44000 Nantes, France; axelle.renodon-corniere@univ-nantes.fr (A.R.-C.); dmitri.levitsky@univ-nantes.fr (D.L.); 2RIKEN Center for Biosystems Dynamics Research, Yokohama 230-0045, Japan; mikawa@riken.jp; 3Structural Biology Research Center, Institute of Materials Structure Science, High Energy Accelerator Research Organization (KEK), Tsukuba 305-0801, Japan; naoyuki.kuwabara@revolka.co.jp; 4Graduate School of Medical Life Science, Yokohama City University, Yokohama 230-0045, Japan; ito.ken.sf@yokohama-cu.ac.jp; 5School of Life Science and Technology, Tokyo Institute of Technology, Tokyo 152-8550, Japan; hiwasaki@bio.titech.ac.jp; 6Innovative Science Institute, Tokyo Institute of Technology, Yokohama 226-8503, Japan

**Keywords:** Rad51 protein, homologous recombination, DNA strand exchange, calcium ion, D-loop

## Abstract

Human Rad51 protein (HsRad51)-promoted DNA strand exchange, a crucial step in homologous recombination, is regulated by proteins and calcium ions. Both the activator protein Swi5/Sfr1 and Ca^2+^ ions stimulate different reaction steps and induce perpendicular DNA base alignment in the presynaptic complex. To investigate the role of base orientation in the strand exchange reaction, we examined the Ca^2+^ concentration dependence of strand exchange activities and structural changes in the presynaptic complex. Our results show that optimal D-loop formation (strand exchange with closed circular DNA) required Ca^2+^ concentrations greater than 5 mM, whereas 1 mM Ca^2+^ was sufficient for strand exchange between two oligonucleotides. Structural changes indicated by increased fluorescence intensity of poly(dεA) (a poly(dA) analog) reached a plateau at 1 mM Ca^2+^. Ca^2+^ > 2 mM was required for saturation of linear dichroism signal intensity at 260 nm, associated with rigid perpendicular DNA base orientation, suggesting a correlation with the stimulation of D-loop formation. Therefore, Ca^2+^ exerts two different effects. Thermal stability measurements suggest that HsRad51 binds two Ca^2+^ ions with K_D_ values of 0.2 and 2.5 mM, implying that one step is stimulated by one Ca^2+^ bond and the other by two Ca^2+^ bonds. Our results indicate parallels between the Mg^2+^ activation of RecA and the Ca^2+^ activation of HsRad51.

## 1. Introduction

The Rad51 protein, a eukaryotic ortholog of the RecA protein, plays a crucial role in homologous recombination by catalyzing strand exchange between two DNAs of identical sequence [1,2]. This function is important for repairing double-strand breaks, resolving stalled replication forks [3], and facilitating chromosome pair formation during meiosis [4]. In vertebrates, the absence of Rad51 is lethal [5]. Rad51 activity is tightly regulated by various proteins in vivo [6,7,8,9]. Correspondingly, in vitro Rad51 activity is affected by these accessory and regulatory proteins [6,7,8,9,10,11,12]. Human and fission yeast Rad51 proteins (HsRad51 and SpRad51) are also stimulated by Ca^2+^ ions in vitro [12,13,14].

Mutations in HsRad51 or regulatory proteins that lead to defects in homologous recombination increase the risk of cancer in humans [15,16,17,18]. Conversely, the overexpression of HsRad51 is frequently observed in human cancer cells and is correlated with poor diagnoses [19,20,21,22]. HsRad51 plays a role in counteracting radiotherapy and chemotherapy by repairing DNA damage caused by these treatments, contributing to the survival of cancer cells [15,21,22,23]. Inhibition of HsRad51 activity has therefore been proposed as a potential cancer treatment [24,25], prompting the development of inhibitors for HsRad51 [25,26,27]. A comprehensive understanding of the molecular mechanisms underlying the strand exchange reaction and the regulation of HsRad51 is crucial for drug development and elucidating the complexity of homologous recombination.

In this study, we examined the mechanism by which Ca^2+^ promotes the stimulation of HsRad51, which is more easily studied than HsRad51 stimulation by a regulatory protein. Ca^2+^ and the activator protein Swi5/Sfr1 exercise similar effects on Rad51, with both promoting the perpendicular orientation of single-stranded DNA (ssDNA) bases in the presynaptic complex [13,28] and stimulating the same reaction step [11,12]. In a previous study, we employed fluorescence resonance energy transfer-based real-time analyses to show that the SpRad51-mediated strand exchange reaction occurs in a three-step process following formation of the presynaptic complex with ssDNA: (1) formation of a three-stranded complex (C1 complex) upon binding of homologous double-stranded DNA (dsDNA) to the presynaptic complex; (2) isomerization of the complex, possibly with topological changes in DNA (C2 complex); and (3) release of the displaced strand from the filament [11,12]. Ca^2+^ and Swi5/Sfr1 stimulate the C1 to C2 transition step.

Perpendicular base alignment may accelerate strand exchange by facilitating the formation of new base pairs (the C1 to C2 transition), however, some differences exist between Ca^2+^ and Swi5/Sfr1. Swi5/Sfr1 also stimulates the third step and ATPase activity of Rad51, whereas Ca^2+^ slows the third step and abolishes ATPase activity [10,11,12,14], suggesting distinct mechanisms of stimulation. ATP hydrolysis by Rad51 is required to ensure the release of the displaced strand from the Rad51 filament [11,12], as well as the release of Rad51 from the newly formed dsDNA [29].

Therefore, in this study, we delved into whether the perpendicular orientation of DNA bases contributes to the stimulation of the strand exchange reaction. To ascertain this, we investigated how the structural transformation of DNA and the enhancement of the strand exchange reaction depend on Ca^2+^ concentration, all under uniform buffer conditions, aiming to identify any correlation.

Strand exchange activity was then assessed using two protocols: strand exchange between (1) two short oligonucleotides and (2) a single-stranded oligonucleotide and a closed circular dsDNA (D-loop formation). The former is widely used for kinetic analyses with fluorescence resonance energy transfer measurements and allows robust assessments [11,12], whereas the latter, which is closer to in vivo conditions, involves topological changes in the closed circular dsDNA.

Subsequently, we examined the Ca^2+^ concentration dependence of structural changes in the presynaptic complex by measuring the linear dichroism (LD) signal and poly(dεA) fluorescence at different Ca^2+^ concentrations. Poly(dεA) serves as a fluorescent analog of poly(dA), with its fluorescence quenched by collisions between εA bases [30]. The intensity increases upon binding of RecA, probably because of the restriction of base movements by RecA [31]. A similar effect is expected upon HsRad51 binding. The LD signal at 260 nm is directly associated with the perpendicular orientation of the DNA bases in the complex [13,28,32]. Given the observed aggregate formation of the presynaptic complex at high Ca^2+^ concentrations, we also examined the Ca^2+^ concentration dependence of the light scattering signals [33,34].

Finally, we examined the binding affinity and stoichiometry through independent measurements and demonstrated Ca^2+^ binding by assessing changes in the thermal stability of HsRad51. Notably, Ca^2+^ binding leads to a decrease in the thermal stability of HsRad51, similar to how Mg^2+^ affects the thermal stability of RecA [35]. Studies on RecA have shown that two Mg^2+^ ions bind to the C-terminal acidic tail of RecA, preventing its interaction with the RecA core, to which DNA binds [35,36,37]. Mg^2+^ thereby relieves the inhibitory effect of the C-terminal tail and stimulates RecA activity. This suggestion is consistent with the destabilization of RecA by Mg^2+^ and the Mg^2+^-independent strand exchange activity of C-terminal tail-deleted RecA [36,37]. We discuss the possible similarity between the activation mechanisms of RecA by Mg^2+^ and HsRad51 by Ca^2+^.

## 2. Results

### 2.1. Optimization of Experimental Conditions

To ensure compatibility among all experiments, we initially established a reaction condition by excluding bovine serum albumin (BSA) from the buffer to avoid interference with circular dichroism (CD) and fluorescence measurements. However, removing BSA led to decreased reproducibility in the strand exchange reaction, possibly due to defects in the surfaces of Eppendorf tubes promoting spontaneous strand exchange without HsRad51. To counteract this, we tested various detergents and surface coating agents and found that adding a low concentration (0.0075% *v*/*v*) of Tween-20 (polyethylene glycol sorbitan monolaurate) significantly improved the reproducibility of the strand exchange reaction and fluorescence measurements. Tween-20 likely prevented Rad51 adsorption to the surfaces of the quartz cuvettes used for fluorescence measurements.

To minimize interference caused by Ca^2+^ binding to free ATP, we reduced the ATP concentration to 300 μM. High ATP concentrations can chelate free Ca^2+^ [38] and interfere with CD measurements owing to strong UV absorption. We verified that 300 μM ATP was sufficient for optimal strand exchange without affecting CD measurements. The strand exchange reaction remained unaffected by ATP concentrations of 100–800 μM when experiments were conducted with 1 mM CaCl_2_. Notably, a slight decrease in strand exchange was observed at 1 mM ATP. This is likely because of a reduction in Ca^2+^ ion availability for HsRad51.

Furthermore, we introduced sodium hydrochloride (NaCl) (50 mM) to mitigate the nonspecific binding of Ca^2+^ to DNA [39]. Experiments were performed without Mg^2+^ to avoid potential complications arising from eventual competition and cooperation between the two divalent ions.

### 2.2. Higher Ca^2+^ Concentration Required for D-Loop Formation than for Oligonucleotide Strand Exchange

We investigated the Ca^2+^ concentration dependence of HsRad51-mediated D-loop formation and strand exchange between two short oligonucleotides. D-loop formation, which involves the strand separation of closed circular DNA and is topologically more constrained than short linear DNA, required a higher concentration of Ca^2+^ than oligonucleotide strand exchange. Oligonucleotide strand exchange was stimulated with approximately 0.3 mM Ca^2+^ for the half-maximum effect, whereas D-loop formation required 2.5 mM Ca^2+^ for the half-maximum effect (Figure 1). These findings suggest that Ca^2+^ has dual effects on HsRad51, and each HsRad51 protomer binds two Ca^2+^ ions.

### 2.3. More than 2 mM of Ca^2+^ Was Required to Saturate the LD Signal

Next, we performed flow LD measurements of the HsRad51-ssDNA-ATP filament at different Ca^2+^ concentrations. In flow LD, a shear force was applied to align the sample molecules, and the difference in absorption between light polarized parallel and perpendicular to the sample orientation axis was measured [32]. Flow LD provides information about the chromophore orientation relative to the filament axis, and the signal intensity depends on the degree of filament orientation itself with respect to its stiffness [32]. Only non-moving chromophores in a stiff filament result in a significant LD signal [32].

The LD spectrum of the HsRad51 presynaptic complex contains information regarding the orientation of all chromophores in the complex, i.e., DNA bases, ATP, and tyrosine residues of HsRad51 [13,28,32]. The signal around 260 nm is dominated by DNA and reflects DNA base alignment. The signals around 230 nm and 280 nm primarily indicate the tyrosine residue orientations of HsRad51 [13,28,32]. Our previous experiments showed that HsRad51 binding to ssDNA in the presence of Mg^2+^ and ATP exhibited an LD signal, revealing the formation of a stiff filament. However, the LD spectrum showed no significant signal from DNA [13,28,32], suggesting local movements or random orientations of the DNA bases. In contrast, Ca^2+^ induced a large negative LD signal from DNA bases, indicating a perpendicular orientation of DNA bases relative to the filament axis in the presence of Ca^2+^ [13].

At various Ca^2+^ concentrations, we measured the LD spectra of the HsRad51/poly(dT)/ATP complex (Figure 2A). No LD signal was observed in the absence of Ca^2+^ (and Mg^2+^), suggesting that stiff filaments were not formed without divalent ions. A negative LD signal at 260 nm appeared at 0.4 mM Ca^2+^ and intensified with increasing Ca^2+^ concentration up to 2 mM (Figure 2A,B). The results showed that 1 mM Ca^2+^ was insufficient to produce a maximum change in LD signal intensity at 260 nm, in contrast with the saturating effect of 1 mM Ca^2+^ on the oligonucleotide strand exchange. The LD signal decreased at 3 mM Ca^2+^ and almost disappeared at 5 mM Ca^2+^, which could be attributed to the aggregate formation (Figure 2B). Estimation of the Ca^2+^ concentration for the half-maximum effect on the LD signal change was, therefore, difficult. However, the requirement of more than 2 mM Ca^2+^ for the maximum signal change suggests that this effect is related to the stimulation of D-loop formation.

A change in the LD signal may result from a modification to the structure or stiffness of the filamentous molecule. The stiffness-related modification will change the intensity of the LD signal without altering the spectral shape. In this study, Ca^2+^ modified the spectral shape. The ratio of the LD signal at 260 nm to that at 230 nm changed with increasing Ca^2+^ concentrations, suggesting that Ca^2+^ alters the structure of the synaptic complex, possibly resulting in better-aligned DNA bases.

The decrease in signal intensity at Ca^2+^ concentrations above 3 mM may be associated with the reduced orientation of the presynaptic filament attributed to aggregate formation. This is supported by the increase in light-scattering-related signals above 300 nm, where no element absorbs.

### 2.4. Saturation of Poly(dεA) Fluorescence Change Requires Less than 1 mM Ca^2+^

To investigate Ca^2+^-promoted structural changes in the presynaptic complex, we measured the fluorescence of the poly(dA) analog, poly(dεA). Fluorescence measurements are less affected by light scattering than LD measurements. The fluorescence intensity of free poly(dεA) is much lower than that of monomeric εA nucleobase because of base/base collisions and stacking [30,31]. The binding of HsRad51 to poly(dεA) increased the fluorescence intensity of poly(dεA) in a Ca^2+^ concentration-dependent manner (Figure 3). The change plateaued at a Ca^2+^ concentration of less than 1 mM. The Ca^2+^ concentration required to reach the half-maximum effect was 0.2 mM, which was much less than that required to change the LD signal.

This implies that HsRad51 binds two Ca^2+^ ions. The binding of one Ca^2+^ ion is sufficient to increase the fluorescence intensity of poly(dεA), which is attributed to a decrease in base/base collisions caused by the restriction of DNA base motion by HsRad51 in the presence of Ca^2+^, whereas the binding of two Ca^2+^ ions is required for the perpendicular alignment of DNA bases.

The Ca^2+^-promoted fluorescence change is related to the activation of HsRad51. Mg^2+^ ions, which result in less activation of HsRad51 [12,14], increased the fluorescence intensity of poly(dεA) to a lesser extent (Figure 3).

### 2.5. Light Scattering Promoted at High Ca^2+^ Concentrations

We subsequently examined if the aggregate state of the presynaptic complex is related to the stimulation of strand exchange by analyzing the aggregate state with light scattering. The light scattering signal increases with molecular size and can be used to detect aggregate formation [33,34]. In this study, light scattering was more pronounced at Ca^2+^ concentrations above 3 mM (Figure 4) and was correlated with the decrease in HsRad51 activation observed at high Ca^2+^ concentrations (Figure 1).

### 2.6. HsRad51 Binds More than Two Ca^2+^ Ions: Thermal Denaturation Results

To determine the binding affinity and stoichiometry of Ca^2+^ ions to HsRad51, we conducted thermal denaturation measurements of HsRad51 at various Ca^2+^ concentrations. Ligand binding often alters protein thermal stability [40], as evidenced by the binding of Mg^2+^, an activating ion for RecA, to RecA, which reduces its thermal stability [35].

In this study, denaturation experiments were performed with no other elements (ATP, DNA) to observe the direct and selective effect of Ca^2+^ on HsRad51. Thermal denaturation was monitored by changes in the CD signal at 222 nm, which reflect the alpha-helix content [41]. HsRad51 exhibited a complex denaturation pattern with three apparent transitions: Tm_1_ = 38 °C, Tm_2_ = 65 °C, and Tm_3_ = 90 °C in the absence of Ca^2+^ (Figure 5A), which may correspond to the independent unfolding of three domains in HsRad51. According to structural analyses of Rad51 by X-ray crystallographic analysis, HsRad51 comprises small N-terminal and C-terminal domains, as well as a large core domain [42,43].

Ca^2+^ reduced the thermal stability of HsRad51 (Figure 5). The reduction of the first transition temperature (Tm_1_) occurred at less than 1 mM Ca^2+^ (Figure 5B,C). The concentration of Ca^2+^ required for the half-maximum effect was approximately 0.2 mM (Figure 5C), similar to the Ca^2+^-promoted change in poly(dεA) fluorescence and the stimulation of oligonucleotide strand exchange. In contrast, changes in the second and third transition temperatures (Tm_2_ and Tm_3_) required higher Ca^2+^ concentrations for saturation (Figure 5A,D). The concentration for the half-maximum effect on the second transition was approximately 2.5 mM (Figure 5D). Because the third transition is broad, precise estimation of the transition temperature (Tm_3_) is difficult. However, rough estimation indicates a similar Ca^2+^ concentration dependence to the second transition (Tm_2_). These results suggest that HsRad51 binds two Ca^2+^ ions per protomer with K_D1_ = 0.2 mM and K_D2_ = 2.5 mM. Stimulation of oligonucleotide strand exchange requires the binding of only one Ca^2+^ ion, whereas D-loop formation requires the binding of two Ca^2+^ ions.

Thermal denaturation of the ATP/Rad51/poly(dT) complex was also affected by Ca^2+^. The first transition was stabilized by Ca^2+^ whereas the second was destabilized, indicating the binding of two Ca^2+^ ions in addition to the Ca^2+^ ion involved in the binding of ATP to HsRad51 (Appendix A).

## 3. Discussion

To elucidate the molecular mechanisms controlling Ca^2+^-mediated activation of HsRad51, we examined HsRad51/Ca^2+^ interactions, Ca^2+^-dependent structural changes in the presynaptic complex, and HsRad51 activities under uniform reaction conditions. Our findings suggest that HsRad51 binds two Ca^2+^ ions. The binding of one Ca^2+^ ion is sufficient to restrict DNA base movements and stimulate strand exchange between two short oligonucleotides. The second Ca^2+^ ion binding further restricts DNA base movement to promote the perpendicular alignment of DNA bases in the presynaptic complex and stimulates D-loop formation. New questions include how Ca^2+^ ion induces the perpendicular alignment of DNA bases and why the formation of a D-loop requires a higher concentration of Ca^2+^. These results also have parallels with Rad51 activation by Swi5/Sfr1 and RecA activation by Mg^2+^ ions [11,12,13,35,36,37].

### 3.1. Similarity with Strand Exchange Activation by Swi5/Sfr1 Protein

Ca^2+^ stimulates the C1 to C2 transition step similar to Swi5/Sfr1 [11,12]. Furthermore, both facilitate the perpendicular orientation of DNA bases in the presynaptic complex [13,28]. However, Ca^2+^ slows the release of the displaced strand whereas Swi5/Sfr1 accelerates it, indicating some difference in their activation mechanism [11,12]. The perpendicular orientation of DNA bases facilitates base pair formation with the complementary strand and accelerates the strand exchange reaction. The different effects on the strand displacement step between Swi5/Sfr1 and Ca^2+^ can be attributed to a different mechanism, possibly related to ATP hydrolysis. Specifically, Swi5/Sfr1 increases the ATPase activity of Rad51, whereas Ca^2+^ inhibits it [14].

The need for higher Ca^2+^ concentrations to stimulate D-loop formation may be attributed to the greater difficulty of inducing base-pair opening in closed circular DNA than in short linear dsDNA [44]. Base-pair opening of closed circular DNA leads to topological constraints. The opening of closed circular DNA likely occurs rarely or for a short period of time. The ssDNA should be ready to pair with it, therefore, a perpendicular base orientation is indispensable. Our preliminary experiments with SpRad51 show that maximal stimulation of the C1 to C2 transition requires more than 5 mM Ca^2+^, supporting the requirement of two Ca^2+^ ions for this step and the link between this step and a topological change in DNA. It would be informative to examine whether the Swi5/Sfr1 concentrations required to stimulate D-loop formation exceed the concentrations required for oligonucleotide strand exchange, similar to our results for Ca^2+^.

### 3.2. Potential Ca^2+^-Binding Site at the N-Terminal Extremity of HsRad51

To better understand the activation mechanism, we first identified potential Ca^2+^-binding sites by locating a cluster of acidic negatively charged residues. In the case of RecA, Mg^2+^ binds to the acidic residue cluster at the C-terminal tail [35,36,37]. We observed a small cluster of acidic residues in the N-terminal end (from residue 8–18) (Figure 6). To confirm this observation, we compared the amino acid sequence of HsRad51 with those of SpRad51 and budding yeast Rad51 (ScRad51). Ca^2+^ stimulates SpRad51 [11,12] but not ScRad51 [14]. Therefore, we expected some difference in the amino acid sequence of the Ca^2+^-binding site between HsRad51 and ScRad51. Their overall sequences are well conserved except for the N-terminal end (Figure 6). Moreover, ScRad51 has a longer N-terminal end than HsRad51 and SpRad51, which supports the hypothesis that Ca^2+^ binds at the N-terminal end of HsRad51. This part is important for the regulation of Rad51. N-terminal acetyltransferase NatB regulates Rad51 [45], and phosphorylation at positions 13 and 14 regulates Rad51 [46]. We prepared a HsRad51 lacking the N-terminal end to test our hypothesis, however, this cluster of acidic residues was too small to bind two Ca^2+^ ions, indicating the need for another Ca^2+^-binding site. Previous research has proposed that Ca^2+^ binding occurs near the ATP-binding site of HsRad51 [47].

### 3.3. Similarity with RecA-Promoted Strand Exchange Activation by Mg^2+^

In parallel with the effect of Mg^2+^ on RecA, Ca^2+^ reduces the thermal stability of HsRad51. Typically, ligand binding increases the thermal stability of proteins [40]. This unusual effect of divalent ions on RecA and HsRad51 implies a common activation mechanism. Mg^2+^ binds to the C-terminal acidic tail of RecA, preventing its interaction with the core of RecA, where DNA binding occurs [35,36,37]. RecA lacking the C-terminal tail becomes insensitive to the Mg^2+^ concentration for thermal stability and strand exchange activity [35,36,37]. Thus, Mg^2+^ alleviates the inhibitory effect of the C-terminal tail and stimulates RecA activity.

It is possible that Ca^2+^ binds at the N-terminal end and prevents its interaction with the DNA-binding site. Interaction of the N-terminal end with the DNA-binding site would hinder the formation of the stable presynaptic complex, with DNA quickly dissociating from and reassociating with Rad51. This unstable binding allows movements of the DNA backbone and bases and slows the formation of new base pairs, which is an important step of the strand exchange reaction. Ca^2+^ may release this inhibitory effect of the N-terminal end and promote a close interaction between DNA and HsRad51. Burgreev and Mazin reported stabilization of the presynaptic complex by Ca^2+^ [14]. Ca^2+^ may strengthen the contact between DNA and HsRad51 and immobilize the DNA bases. The increase in fluorescence intensity of poly(dεA) and the strong negative LD signal at 260 nm by Ca^2+^ showed the restriction of DNA base movements. The binding of one Ca^2+^ may not be sufficient to completely stabilize the presynaptic complex and immobilize the DNA bases.

We investigate the validity of this assumption through experiments and molecular simulations. Notably, the Swi5/Sfr1 dimer also stabilizes the presynaptic complex by slowing the dissociation rate [11,12]. Although Ca^2+^ may not play a regulatory role in vivo [48], understanding its molecular mechanism will be useful to clarify the action of biological regulatory elements.

## 4. Materials and Methods

### 4.1. Materials

Poly(dT), poly(dA), ATP, and Tween-20 were obtained from Sigma-Aldrich (St. Louis, MO, USA). HsRad51 was purified according to a previously described procedure [49]. Poly(dεA) was synthesized by chemical modification of poly(dA) using chloroacetaldehyde (Aldrich), according to a previous method [50]. The degree of modification was approximately 93%, determined by spectroscopy using the formula provided by Ledneva et al. [51]. Concentrations were determined based on UV absorption, utilizing the following extinction coefficients: ε_263nm_ = 8520 M^−1^ cm^−1^ for poly(dT), ε_257nm_ = 3800 M^−1^ cm^−1^ for poly(dεA), and ε_260nm_ = 15,400 M^−1^ cm^−1^ for ATP.

### 4.2. Experimental Conditions

Experiments were performed in a buffer containing 30 mM Tris/HCl (pH 7.5), 50 mM NaCl, 0.1 mM ethylenediamine tetraacetic acid, 0.0075% Tween-20, and the specified concentrations of Ca^2+^. The ATP concentration, if present, was maintained at 300 μM. In this study, the concentration of free Ca^2+^ (after deduction of 0.1 mM Ca^2+^ chelated by ethylenediamine tetraacetic acid) is given without additional mention. Tween-20 was added to prevent the adsorption of HsRad51 onto the surface of the quartz cell or plastic tube. The concentration of HsRad51 was 2 μM except for the strand exchange and D-loop formation experiments (0.5 μM).

### 4.3. LD Measurements

LD spectra were recorded with a Jasco J-810 CD spectrometer in step mode (data interval: 0.1 nm; time constant: 0.175 s; bandwidth: 2 nm) and LD mode. The samples were aligned using a mini-Couette cell (Jasco Europe, Cremella, Italy), and the baseline was determined by slowly rotating the Couette cell filled with the sample to average the defects in the cell.

### 4.4. CD Measurements of Thermal Unfolding

Thermal unfolding of HsRad51 was monitored by tracking the change in the CD signal at 222 nm (bandwidth, 5 nm; data interval, 0.1 °C; time constant, 2 s) with increasing temperature (1 °C/min). A mini cuvette of 1 × 0.2 cm (Hellma GmbH & Co., Müllheim, Germany) with a path length of 1 cm was used for the measurements. The temperature was controlled with a Peltier effect controller.

### 4.5. Fluorescence Measurements

The fluorescence of poly(dεA) was measured with a Jasco FP-8300 fluorometer (JASCO Corporation, Hachioji, Japan. The emission signal at 380 nm (bandwidth: 10 nm) was recorded upon excitation at 320 nm (bandwidth: 5 nm). The results from 20 measurements were averaged, and the temperature was maintained at 20 °C using a Peltier effect controller.

### 4.6. Light Scattering Measurements

Scattered light at 90° to the incident light was measured in a Jasco FP-8300 fluorometer. Light of 400 nm (bandwidth 10 nm) was used. The results from 20 measurements were averaged, and the temperature was maintained at 20 °C using a Peltier effect controller. Dust in the solution was eliminated by mild centrifugation.

### 4.7. Strand Exchange and D-Loop Formation

DNA strand exchange between two short oligonucleotides was performed as described previously [28], except BSA was omitted. Briefly, 59-mer* ssDNA (1.16 µM) was incubated at 37 °C for 1 h with HsRad51 (0.5 µM) and 32-mer dsDNA (1.65 µM bp) in the presence of indicated concentrations of Ca^2+^. The reactions were stopped and deproteinized by the incubation of SDS (0.7%) and proteinase K (0.7 mg/mL) for 15 min at 37 °C. The reaction products were separated by electrophoresis on 15% polyacrylamide gel.

D-loop formation experiments were performed as described in a previous study [14] but under the buffer conditions of this study. Briefly, 100-mer* ssDNA (1 µM) was incubated at 37 °C for 30 min with HsRad51 (0.5 µM) and supercoiled plasmid pPB4.3 DNA (200 µM bp) in the presence of indicated concentrations of Ca^2+^. The reactions were stopped and deproteinized by the incubation of SDS (1%) and proteinase K (1 mg/mL) for 15 min at 37 °C. The reaction products were separated by electrophoresis on 1% agarose gel.

The following oligonucleotides were used: 32-mer 5′-CCA TCC GCA AAA ATG ACC TCT TAT CAA AAG GA-3′; 32-mer 5′-TCC TTT TGA TAA GAG GTC ATT TTT GCG GAT GG-3′; 59-mer* 5′-TCC TTT TGA TAA GAG GTC ATT TTT GCG GAT GGC TTA GAG CTT AAT TGC TGA ATC TGG TG-3′; 100-mer* 5′-GGG CGA ATT GGG CCC GAC GTC GCA TGC TCC TCT AGA CTC GAG GAA TTC GGT ACC CCG GGT TCG AAA TCG ATA AGC TTA CAG TCT CCA TTT AAA GGA CAA G-3′. The 5′ end of 59-mer* and 100-mer* were labeled with IRD dye. The labeled products were visualized and quantified by the detection of the IRD dye with the infrared fluorescent detection channel of an Odyssey Infrared Imager (LI-COR).

## Figures and Tables

**Figure 1 ijms-25-03633-f001:**
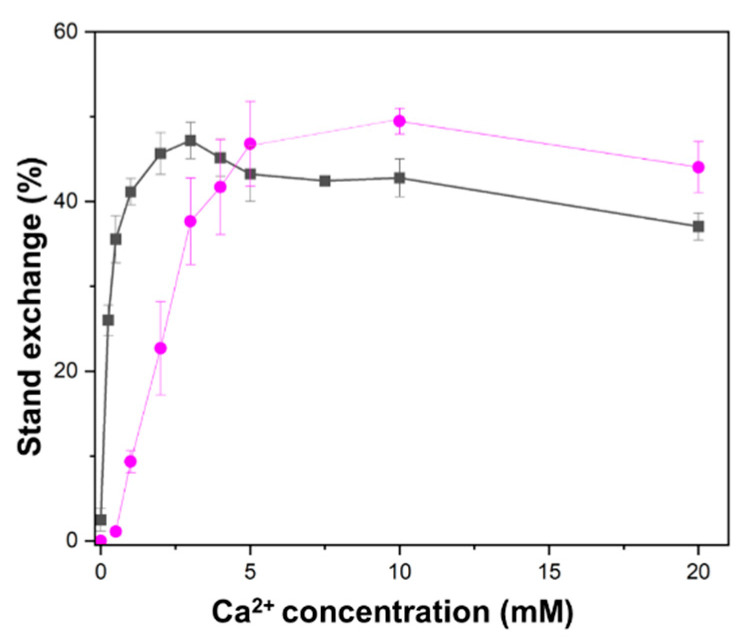
Differential Ca^2+^ concentration dependence between D-loop formation and oligonucleotide strand exchange. HsRad51-mediated D-loop formation (magenta) and strand exchange between two short oligonucleotides (black) were measured at various Ca^2+^ concentrations.

**Figure 2 ijms-25-03633-f002:**
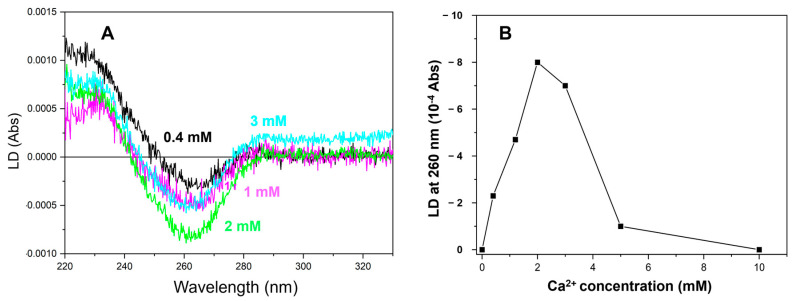
LD spectra of ATP-poly(dT)-HsRad51 complex at various Ca^2+^ concentrations. (**A**) LD spectra of the ATP-poly(dT)-HsRad51 complex recorded at various Ca^2+^ concentrations (shown by different colors). (**B**) LD signal intensity at 260 nm plotted as function of Ca^2+^ concentration. Since a baseline shift was observed for the signal at 3 mM Ca^2+^, signal intensity was corrected. Such correction was not necessary for the signal obtained with other Ca^2+^ concentrations due to absence of baseline shift.

**Figure 3 ijms-25-03633-f003:**
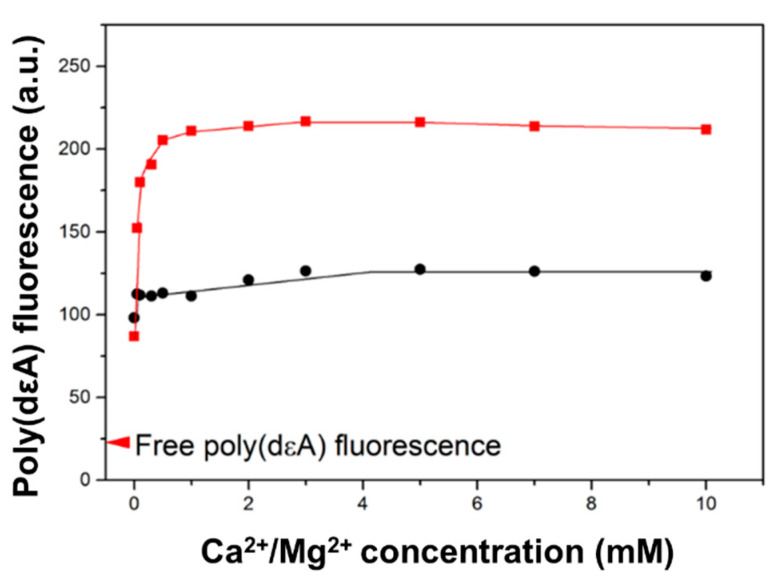
Effect of Ca^2+^ on poly(dεA) fluorescence intensity. Fluorescence intensity of poly(dεA) in HsRad51-ATP filament is shown at various Ca^2+^ (red symbols) and Mg^2+^ (black symbols) concentrations.

**Figure 4 ijms-25-03633-f004:**
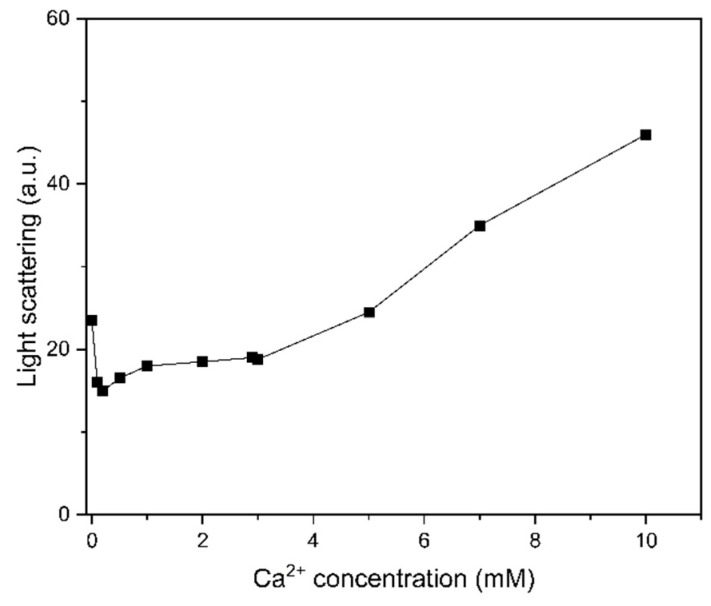
Light scattering of the ATP-poly(dT)-HsRad51 complex at various Ca^2+^ concentrations measured at 400 nm. Relative signal intensity was plotted as function of Ca^2+^ concentration.

**Figure 5 ijms-25-03633-f005:**
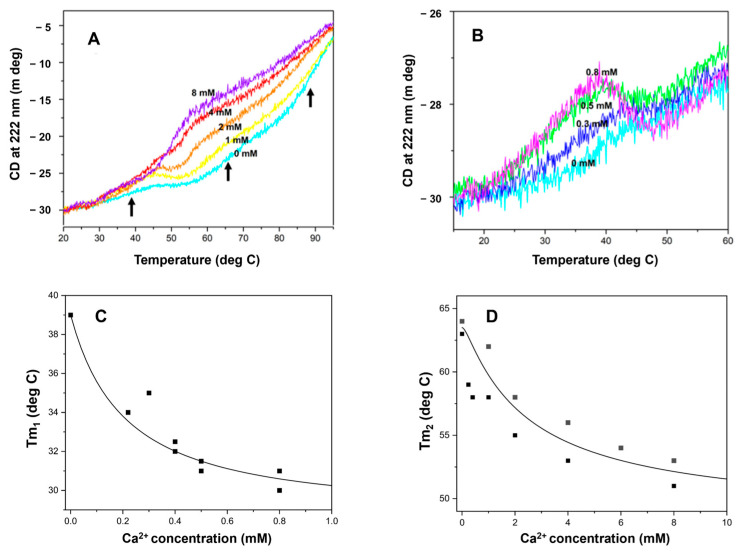
Ca^2+^ concentration dependence of HsRad51 thermal denaturation. (**A**) Ca^2+^ binding of HsRad51 evaluated by measuring thermal denaturation at different Ca^2+^ concentrations (shown by different colors). Denaturation was monitored by measuring changes in CD signal at 222 nm under increasing temperature. Arrows indicate three transitions. (**B**) Detailed changes in denaturation profile with increasing Ca^2+^ concentration (0–0.8 mM). (**C**,**D**) Change in Tm of first transition (Tm_1_, (**C**)) and second transition (Tm_2_, (**D**)) with Ca^2+^ concentration. A theoretical curve with K_D1_ = 0.2 mM and K_D2_ = 2.5 mM is also shown. Each symbol corresponds to result of one independent experiment.

**Figure 6 ijms-25-03633-f006:**
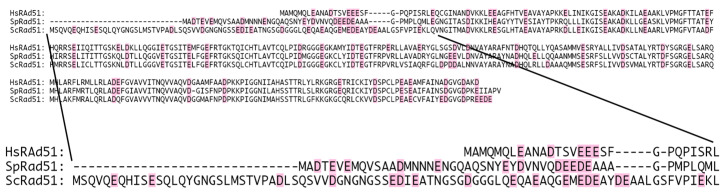
Cluster of negatively charged amino acid residues at N-terminal of HsRad51. Amino acid sequence of HsRad51 is compared with amino acide sequence of SpRad51 and ScRad51. Negatively charged residues are indicated in magenta.

## Data Availability

Data are contained within the article and Appendix A.

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
