# Peer review of "Human Rad51 Protein Requires Higher Concentrations of Calcium Ions for D-Loop Formation than for Oligonucleotide Strand Exchange"

_ijms, 2024, doi:10.3390/ijms25073633_

Round 1

Reviewer 1 Report

Comments and Suggestions for Authors

The manuscript, “Human Rad51 protein requires higher concentrations of calcium ions for D-loop formation than for oligonucleotide strand exchange” delves into the calcium dependent mechanism in human Rad51’s DNA strand exchange using multiple analytical techniques. While the exploration employs diverse DNA structures and various calcium concentrations, significant improvements are needed before publication.

The manuscript did not highlight the critical role of ATPase activity in ssDNA binding by Rad51 and its influence on homologous recombination. While mentioned briefly and included ATP concentration optimization, a detailed description and analysis of ATPase activity across calcium concentrations and DNA structures is necessary.

The observation of destabilization at higher calcium concentrations could be non-specific based on previous reports, 1) Rad51 is known to bind to calcium in the presence of ATP (Louise H. et al., Nucleic Acids Research, Volume 40, Issue 11, 1 June 2012, Pages 4904–4913) and 2) there are crystal structures of Rad51 in the presence of both calcium and ATP (ex, Appleby, R. et al., iScience, 26 (5), 2023,106689). Have other cations been tested as controls?

The description between lines 69-101are more appropriate for the result section.

Some minor points:

Figure Legend Placement: Correct the positioning of legends for Figures 1, 3, and 5.

FRET Relevance: If no FRET experiments were conducted, remove lines 54-60 in the Introduction for clarity.

Data Point Clarification: Explain the presence of two data points at the same calcium concentrations in Figures 2C and 2D.

Method: Provide detailed description, provide sequence information for single-strand oligonucleotides in Figure 1.and include light scattering experiment

Author Response

Referee 1.

The manuscript, “Human Rad51 protein requires higher concentrations of calcium ions for D-loop formation than for oligonucleotide strand exchange” delves into the calcium dependent mechanism in human Rad51’s DNA strand exchange using multiple analytical techniques. While the exploration employs diverse DNA structures and various calcium concentrations, significant improvements are needed before publication.

The manuscript did not highlight the critical role of ATPase activity in ssDNA binding by Rad51 and its influence on homologous recombination. While mentioned briefly and included ATP concentration optimization, a detailed description and analysis of ATPase activity across calcium concentrations and DNA structures is necessary.

In the revised introduction section, we refer to the important roles of ATP hydrolysis in the strand exchange reaction. We also mention that Ca2+ ions abolish the ATP hydrolysis activity of HsRad51.

The observation of destabilization at higher calcium concentrations could be non-specific based on previous reports, 1) Rad51 is known to bind to calcium in the presence of ATP (Louise H. et al., Nucleic Acids Research, Volume 40, Issue 11, 1 June 2012, Pages 4904–4913) and 2) there are crystal structures of Rad51 in the presence of both calcium and ATP (ex, Appleby, R. et al., iScience, 26 (5), 2023,106689). Have other cations been tested as controls?

We appreciate the reviewer’s insights regarding the involvement of one Ca2+ ion in the binding of ATP to Rad51 and the iScience paper. We have noted in the revised manuscript that Rad51 binds two Ca2+ ions in addition to Ca2+ for ATP binding.

Ca2+ also decreases the thermal stability of the ATP-Rad51-poly(dT) complex. We have added a description of this result to the supplementary material.

The description between lines 69-101 are more appropriate for the result section.

The intent of including this text was to show the flow of the experiments. However, we have deleted sentences that were more appropriate to the results section.

Some minor points:

Figure Legend Placement: Correct the positioning of legends for Figures 1, 3, and 5.

We have corrected the placement of figure legends throughout the manuscript.

FRET Relevance: If no FRET experiments were conducted, remove lines 54-60 in the Introduction for clarity.

To avoid confusion, we have clarified that this text is referring to previously reported results:

“In a previous study, we employed fluorescence resonance energy transfer-based real-time analyses to show that the SpRad51-mediated strand exchange reaction occurs in a three-step process following formation of the presynaptic complex with ssDNA”

Data Point Clarification: Explain the presence of two data points at the same calcium concentrations in Figures 2C and 2D.

The presence of two data points is because the experiments were performed twice. We have explained this in the figure legend.

Method: Provide detailed description, provide sequence information for single-strand oligonucleotides in Figure 1.and include light scattering experiment

We have added the relevant methodological descriptions to the materials and methods section.

Reviewer 2 Report

Comments and Suggestions for Authors

Please see the attached file for review comments.

Author Response

The topic of the article “Human Rad51 protein requires higher concentrations of calcium ions for D-loop formation than for oligonucleotide strand exchange” is interesting, however some concerns need to be addressed before the manuscript is ready for publication in the journal IJMS.

General comments:

  1. The presentation of results should be improved.

We have modified the flow and structure of the results section and amended the figures.

  1. Titles of sections and subsections should be numbered.

We have numbered all section and subsection headings.

  1. Fonts used for the sections’ titles should be unified.

A consistent font has been used throughout the manuscript.

  1. The References section should be carefully corrected.

We have corrected the reference format.

  1. The order of authors in the manuscript file is different than that indicated in the Review Report Form (Masayuki Takahashi as first vs. last author). This issue should be addressed.

We have corrected the order of authors in the Review Report Form. The order in the manuscript file is correct.

Specific comments:

  1. Line 32: The green highlighting of semicolons should be removed.

The green highlighting is not present in the original file. It may have appeared during transformation of the Word file into the submission format.

  1. Line 78: I suggest to consider “by independent measurements” instead of “by independent measurement”.

Thank you for your suggestion; we have modified the text accordingly.

  1. Line 91: I suggest “two Ca2+ ions. In contrast, one Ca2+ ion is” instead of “two Ca2+. In contrast,

one Ca2+ is”.

We have deleted this part in accordance with reviewer 1’s suggestion.

  1. Line 96: I suggest “Mg2+ ion binds” instead of “Mg2+ binds”.
  2. Line 110: It should be “(0.0075% v/v)” instead of “(0.0075% V/V)”.
  3. Page 4, Figure 1, X-axis title: It should be “Ca2+ (mM) instead of “Ca2+ (mM)”.

We have made the above suggested corrections in the revised manuscript.

  1. Lines 139-140: The sentence “HsRad51-mediated D-loop formation (magenta) and strand exchange between two short oligonucleotides (black) were measured at various Ca2+ concentrations” should be moved above as a continuation of the Figure 1 caption (lines 137-138).

This error occurred during transformation of the Word file to the submission format. We have attempted to ensure that this sentence appears in the correct location.

  1. Line 155: An incorrect ion. It should be “1 mM Ca2+ (Figs. 2B” instead of “1 mM Mg2+ (Figs. 2B”.

We have made the suggested corrections in the revised manuscript.

  1. Lines 162 and 163: I suggest “one Ca2+ ion,” and “two Ca2+ ions.” instead of “one Ca2+,” and “two Ca2+.”, respectively.

We have made the suggested corrections in the revised manuscript.

  1. Page 5, Figure 2B, X-axis title: It should be “Temperature (deg C)” instead of “Temperature (C)”.

We have made the suggested corrections in the revised manuscript.

Please see Figure 2A for comparison.

  1. Page 5, Figures 2C and 2D, X-axis title: It should be “Ca2+ concentration (mM)” instead of “Ca2+ concentration (mM)”.

We have made the suggested corrections in the revised manuscript.

  1. Page 5, Figures 2C and 2D, Y-axis title: Subscript fonts should be used in symbols of transition temperatures (please see also comment no. 20).

We have made the suggested corrections in the revised manuscript.

  1. Lines 167-168, Figure 2 caption: It should be “at 222 nm” instead of “at 220 nm”.

We have made the suggested corrections in the revised manuscript.

  1. Lines 168 and 170, Figure 2 caption: An unnecessary space?

We have made the suggested corrections in the revised manuscript.

  1. Lines 170-173, Figure 2 caption: Please verify description for panels (C) and (D). Moreover, the statement “(D) Change in the Tm of the third transition (Tm3) with Ca2+ concentrations.” is not consistent with the Y-axis title in Figure 2D “Tm2 (deg C)” (please see also lines 159-160 for comparison as well as comment no.17).

We have corrected the error so that both the caption and y-axis title refer to the change in the Tm of the third transition (Tm2) with Ca2+ concentration.

  1. Page 6, Figure 3, X-axis title: It should be “Ca2+/Mg2+ concentration (mM)” instead of “Ca/Mg concentration (mM)”.

This has been corrected.

  1. Page 6, Figure 3, Y-axis title: It should be ”poly(dεA) fluorescence” instead of ”poly(dA) fluorescence”. Shouldn't units for fluorescence be given?

We have made the necessary correction and expressed the fluorescence intensity in relative units (a.u.).

  1. Lines 189-190: The sentence “The fluorescence intensity of poly(dεA) in the HsRad51-ATP filament was measured at various Ca2+ (red symbols) or Mg2+ concentrations (black symbols).” should be moved above as a continuation of the Figure 3 caption (line 188).

This error occurred during transformation of the Word file to the submission format.

  1. Line 221: I suggest “two Ca2+ ions” instead of “2 Ca2+ ions”.

We have made the suggested corrections in the revised manuscript.

  1. Page 7, Figure 4B: Please compare the value of the LD signal intensity at 260 nm given in Figure 4B for 3 mM Ca2+ with that presented in Figure 4A for the same Ca2+ concentration (it seems that the value for 3 mM Ca2+ in Figure 4B is not consistent with that given in Figure 4A).

For the signal at 3 mM Ca2+, the intensity was corrected for the deviation of the baseline due to aggregation formation. This is mentioned in the figure caption.

  1. Line 224, Figure 4 caption: It should be ”(A)” instead of “(a)”.

Again, this error occurred during transformation of the Word file to the submission format.

  1. Page 8, Figure 5: Is it possible to have Ca2+ concentrations below 0 mM? This issue should be addressed.

The Ca2+ concentration was miscalculated. We have amended this error and prepared a new figure.

  1. Lines 246-248: The sentence “Light scattering of the ATP-poly(dT)-HsRad51 complex with various Ca2+ concentrations was measured at 400 nm. Relative signal intensity was plotted as a function of Ca2+ concentration” should be moved above as a continuation of the Figure 5 caption (line 245).

We have made the suggested corrections in the revised manuscript.

  1. Line 254: It should be “two Ca2+ ions” instead of “two Ca2+ molecules”.

We have made the suggested corrections in the revised manuscript.

  1. Lines 254-255: I suggest “one Ca2+ ion is sufficient” instead of “one Ca2+ is sufficient”.

We have made the suggested corrections in the revised manuscript.

  1. Line 256: I suggest “The second Ca2+ ion binding stimulated” instead of “The second Ca2+ binding stimulated” (please see the comment above for comparison).

We have made the suggested corrections in the revised manuscript.

  1. Lines 259-260: I suggest to add the relevant reference number at the end of the sentence.

We have made the suggested additions in the revised manuscript.

  1. Lines 260 and 281: I suggest “Swi5-Sfr1” instead of “Swi5/Sfr1” (please see lines 18, 61, 64, 261, 263, 269, 270, and 326 for comparison).

We have made the suggested corrections in the revised manuscript.

  1. Lines 263-264: I suggest to verify the statement ”Swi5-Sfr1 accelerates the release of the displaced strand release and the transition from C1 to C2”.

We have made the suggested corrections in the revised manuscript.

  1. Lines 278-279: I suggest “requirement of two Ca2+ ions for“ instead of “requirement of two Ca2+ for“.

We have made the suggested corrections in the revised manuscript.

  1. Line 286: An unnecessary space?

We have made the suggested corrections in the revised manuscript.

  1. Lines 301-302: The sentence “The amino acid sequence of HsRad51 is shown in comparison with that of SpRad51 and ScRad51. Negatively charged residues are indicated in magenta.” should be moved above as a continuation of the Figure 6 caption (line 300).

We have moved the sentence to the suggested position.

  1. Line 322: I suggest “The binding of one Ca2+ ion may” instead of “The binding of one Ca2+ may”.

We have made the suggested corrections in the revised manuscript.

  1. Line 327: I suggest “in vivo” instead of “in vivo” (please see lines 41 and 74 for comparison).

We have made the suggested corrections in the revised manuscript.

  1. Lines 340-341: I suggest “Tris/HCl (pH 7.5)” instead of “Tris/hydrochloric acid (pH 7.5)”.

We have made the suggested corrections in the revised manuscript.

  1. Line 361: A necessary space is missing. It should be ”20 °C” instead of “20°C”.

We have made the suggested corrections in the revised manuscript.

  1. The References section: Authors names should be provided in a consistent manner – please see Refs. 15, 16, 19, 21, 37, 45.

We have made the necessary corrections.43. The References section: Journal names should be written in a uniform manner (full/abbreviated). When abbreviated, journal names should be unified (with or without punctuation marks).

We have made the suggested corrections in the revised manuscript.

  1. The References section: I suggest to write titles of articles in the uniform manner (upper/lower case).

We have made the suggested corrections in the revised manuscript.

  1. Line 449, Ref. 37: It should be “Mg2+” instead of “Mg2+”.

We have made the suggested corrections in the revised manuscript.

  1. Line 470, Ref. 49: It should be “Saccharomyces cerevisiae” instead of “Saccharomyces cerevisiae”. Italic fonts should be used for Latin names.

We have made the suggested corrections in the revised manuscript.

Round 2

Reviewer 1 Report

Comments and Suggestions for Authors

The authors have made a commendable effort to address the majority of comments and provide additional information. However, a few revisions are still necessary for publication.

Lines 235-239: The description of a three-step transition with Ca2+ binding suggests that hsRad51 might bind more than two Ca2+ ions although the estimation of Tm3 (third transition temperature) seems impractical. Please clarify this section.

Lines 173-174: The manuscript mentions data correction at 3mM Ca2+ concentration. Please explain the rationale and method used for this correction, and why other data points were not corrected similarly.

Protein Concentration: Please ensure consistent reporting of hsRad51 concentration throughout all experiments described in the manuscript.

Section 4.7: For clarity, please provide a brief description of the experimental procedures used in section 4.7.

Author Response

The authors have made a commendable effort to address the majority of comments and provide additional information.

However, a few revisions are still necessary for publication.

Lines 235-239: The description of a three-step transition with Ca2+ binding suggests that hsRad51 might bind more than two Ca2+ ions although the estimation of Tm3 (third transition temperature) seems impractical. Please clarify this section.

Thank you for this valuable remark. As you mentioned, precise estimation of Tm3 is difficult because of its broad transition. However, rough estimation indicates similar Ca2+ dependence as Tm2. We have explained this in the text.

Lines 173-174: The manuscript mentions data correction at 3 mM Ca2+ concentration. Please explain the rationale and method used for this correction, and why other data points were not corrected similarly.

We are sorry that the explanation was confusing. The correction is required only for 3 mM Ca2+ because only the LD spectrum with 3 mM Ca2+ presents a baseline shift (the signal 300-350 nm was not zero).  In contrast, the spectra with other Ca2+ concentrations do not present such a shift. Therefore, the correction is not required. We explain this in the text.

Protein Concentration: Please ensure consistent reporting of hsRad51 concentration throughout all experiments described in the manuscript.

We described the concentration of HsRad51 in “Materials and methods”.

Section 4.7: For clarity, please provide a brief description of the experimental procedures used in section 4.7.

We describe the procedures in more detail as:

“DNA strand exchange between two short oligonucleotides was performed as described previously [28], except BSA was omitted. Briefly, 59-mer* ssDNA (1.16 µM) was incubated at 37 °C for 1 h with HsRad51 (0.5 µM) and 32-mer dsDNA (1.65 µM bp) in the presence of indicated concentrations of Ca2+. The reactions were stopped and deproteinized by the incubation of SDS (0.7%) and proteinase K (0.7 mg/mL) for 15 min at 37 °C. The reaction products were separated by electrophoresis on 15% polyacrylamide gel.

D-loop formation experiments were performed as described in a previous study [14] but under the buffer conditions of this study. Briefly, 100-mer* ssDNA (1 µM) was incubated at 37 °C for 30 min with HsRad51 (0.5 µM) and supercoiled plasmid pPB4.3 DNA (200 µM bp) in the presence of indicated concentrations of Ca2+. The reactions were stopped and deproteinized by the incubation of SDS (1 %) and proteinase K (1 mg/mL) for 15 min at 37 °C. The reaction products were separated by electrophoresis on 1% agarose gel.

The following oligonucleotides were used: 32-mer 5′-CCA TCC GCA AAA ATG ACC TCT TAT CAA AAG GA-3′; 32-mer 5′-TCC TTT TGA TAA GAG GTC ATT TTT GCG GAT GG-3′; 59-mer* 5′-TCC TTT TGA TAA GAG GTC ATT TTT GCG GAT GGC TTA GAG CTT AAT TGC TGA ATC TGG TG-3′; 100-mer* 5’ -GGG CGA ATT GGG CCC GAC GTC GCA TGC TCC TCT AGA CTC GAG GAA TTC GGT ACC CCG GGT TCG AAA TCG ATA AGC TTA CAG TCT CCA TTT AAA GGA CAA G-3’. The 5’ end of 59-mer* and 100-mer* were labeled with IRD dye. The labeled products were visualized and quantified by the detection of the IRD dye with the infrared fluorescent detection channel of an Odyssey Infrared Imager (LI-COR).”